# Laser-Induced Ion Formation and Electron Emission from a Nanostructured Gold Surface at Laser Fluence below the Threshold for Plasma Formation

**DOI:** 10.3390/nano13030600

**Published:** 2023-02-02

**Authors:** Andrey Pento, Ilya Kuzmin, Viacheslav Kozlovskiy, Lei Li, Polina Laptinskaya, Yaroslav Simanovsky, Boris Sartakov, Sergey Nikiforov

**Affiliations:** 1Prokhorov General Physics Institute of the Russian Academy of Sciences, 119991 Moscow, Russia; 2Vernadsky Institute of Geochemistry and Analytical Chemistry of the Russian Academy of Sciences, 119991 Moscow, Russia; 3Chernogolovka Branch of the N.N. Semenov Federal Research Center for Chemical Physics, Russian Academy of Sciences, Chernogolovka, 142432 Moscow Region, Russia; 4Institute of Mass Spectrometry and Atmospheric Environment, Jinan University, Guangzhou 510632, China; 5Guangdong Provincial Engineering Research Center for On-Line Source Apportionment System of Air Pollution, Guangzhou 510632, China

**Keywords:** gold nanoparticles, laser mass spectrometry, laser-induced nanostructures

## Abstract

The laser formation of positive and negative ions on a nanostructured metal surface is observed at laser fluence below the plasma formation threshold. The laser radiation energy dependences of the yield of positive and negative Au ions and charged clusters as well as electrons from the laser-induced nanostructures on the surface of gold are obtained at laser fluence below the plasma formation threshold using a pulsed laser with a wavelength of 355 nm and a pulse duration of 0.37 ns. It is shown that the ratio of the signals of positive and negative ions is constant over the entire range of the laser radiation energies, while the ion signal dependence on the laser radiation energy is described by a power function with an exponent of 9. The role of gold nanoparticles with a size of less than 5 nm in the formation of Au ions and charged Au clusters is discussed.

## 1. Introduction

The processes resulting in the formation of ions on a metal surface under exposure to pulsed laser radiation have been studied almost since the development of pulsed lasers. The main mechanism leading to the appearance of an intense flow of ions of the target material exposed to high-power laser radiation is the formation of plasma above the target surface. This effect underlies laser breakdown (spark) spectroscopy, an efficient technique that allows one to determine the elemental composition of a sample [1,2], and laser mass spectrometry using laser-induced plasma, which is also aims to analyze the elemental composition [3].

Plasma formation is usually considered to be a threshold process in terms of the energy density or laser radiation intensity. The plasma generation threshold depends on the radiation parameters, the reflection and absorption coefficient of the metal, and its thermophysical properties. Plasma formation requires the effective absorption of laser radiation by the target material, as a result of which a stream of evaporated matter is formed, wherein atoms are ionized by electrons accelerated in the laser field. This is an avalanche process leading to the strong dependence of the plasma parameters on the energy density of the laser radiation. Numerous experiments performed mainly with Nd:YAG laser fundamental radiation and its harmonics have shown that, with a pulse duration of several nanoseconds, the plasma formation threshold for various metals is within the range of 1–10 J/cm^2^ [4].

At the same time, it was found [5] that the appearance of metal ions from a target is detected at a laser energy (power) density lower than the threshold energy density of plasma generation. In this work, using a time-of-flight mass spectrometer, it was experimentally shown that the appearance of a flux of positive metal ions under the action of pulsed radiation with a wavelength of 532 nm also has a threshold character, and the threshold for the appearance of ions is 1.2–1.5 times lower than the threshold for the appearance of plasma.

The threshold for the appearance of metal ions depends on the state of the surface. It was shown [6] that the threshold for the appearance of ions from the nanostructured surfaces of gold, silver, platinum, and copper made via processing with sandpaper decreases to 10 mJ/cm^2^ when the surface is exposed to pulsed laser radiation (532 nm, 4 ns). This effect, in the opinion of the authors, is associated with the resonance excitation of surface plasmons. At the same time, the dependence of the yield of ions near the threshold has not been measured, while an exact comparison of the yields of positive and negative ions has not been performed.

Nanostructures on the surface of a metal can be created not only by mechanical processing, but also via exposure to powerful pulsed radiation. The impact of high-power pulsed radiation leads to the formation of a cloud of nanoparticles near the surface [7] as well as the appearance of various nanostructures on the metal surface with a size range from a few nanometers to several micrometers [8,9,10]. The laser nanostructuring of surfaces and/or laser “activation” also lowers the ion formation threshold and changes the ratio between the signal of Au^+^ ions and the signal of Au_n_^+^ (n ≥ 2) ions [11]. The decrease in the ion formation threshold on a nanostructured surface occurs due to a change in the mechanism of interaction of the laser radiation with the metal surface. An essential role in this process should be played by the excitation of the electronic transitions in nanoparticles of a few nanometers in size, which behave like quantum dots.

The study of ion formation on nanostructured surfaces is important both for understanding the processes of interaction between radiation and matter and for a number of practical applications. An example of such an application is the detection of single aerosol particles in a laser mass spectrometer [12,13]. In such a device, particles enter a laser field of different intensity, which is due to the design of the device. Understanding the processes of ion formation at laser fluence above and below the plasma formation threshold would allow one to correctly interpret the resulting mass spectra.

Finally, various nanoparticles, including gold nanoparticles, are used in the surface-assisted laser desorption ionization mass spectrometry (SALDI-MS) of organic compounds [14]. In this method, ions of organic compounds are formed under the pulsed UV laser irradiation of a mixture of analytes and nanoparticles. Along with the ions of organic compounds under study, the ions of the material of the nanoparticles (gold or other metals) are observed in the mass spectra [15]. This process may be due to the excitation and destruction of the nanoparticles under exposure to pulsed UV radiation, which is similar to the effect observed upon irradiation with picosecond laser pulses [16].

The aim of this work is to study the processes of positive and negative Au ions formation when a nanostructured gold surface is exposed to pulsed laser radiation with a wavelength of 355 nm and a pulse duration of 0.37 ns at laser fluence below the plasma formation threshold. A specially designed time-of-flight mass spectrometer was used to analyze the ions. The use of a mass spectrometer requires the use of a clean nanostructured surface, as the presence of organic impurities that arise, for example, when nanoparticles are deposited onto a metal surface from a solution or during the sorption of volatile organic compounds from air, leads to a distortion of the mass spectra and makes their interpretation more challenging and complicated. To prepare the nanostructured gold surface, we used the laser ablation of the metal in a vacuum. Both the surface preparation and subsequent mass spectrometric analysis were carried out in the mass spectrometer chamber using the same pulsed laser but at different laser fluence. This approach makes it possible to obtain and study a clean nanostructured metal surface.

The complexity of carrying out such measurements is related to the very low probability of ion formation during a particular laser pulse at the lowest laser radiation density. To obtain stable results, we used an accumulation of a large number of mass spectra. However, during irradiation, the parameters of the nanostructures on the metal surface change, which leads to the distortion of the results. To overcome this limitation, we used the scanning of the laser beam over a nanostructured metal surface. Scanning allows for the repeated irradiation of an untouched nanostructured surface.

We also carried out experiments involving multiple repeated irradiation of the surface at a fixed point and compared the results.

## 2. Materials and Methods

### 2.1. Mass Spectrometric Measurements

The experimental arrangement is shown schematically in Figure 1. A 0.5 mm thick gold plate (2) was installed in the chamber of a linear time-of-flight mass spectrometer (1) equipped with windows for the laser radiation input and the observation of the studied surface with a CCD camera. The pressure in the chamber was 3 × 10^−7^ Torr. The mass resolution of the instrument was about 500 for *m*/*z* 100. Only one grid was installed in the ion path, which ensured that the ion transmission coefficient was no less than 0.5. The ions were registered with an F-9892-11 microchannel plate chevron assembly (Hamamatsu Photonics K.K., Hamamatsu City, Japan) (4). The laser radiation passed through a reflection-type variable attenuator (7) and a 2D scanner (9), and it was focused on the metal surface by a plano-convex spherical lens (8) with a focal length of 50 cm. The Au ion yield was measured in two different modes: in static mode, when the laser beam was directed to a fixed point on the metal surface, and in scanning mode, when the beam was serpentine scanned over a fixed area on the surface.

The mass spectra were recorded using an AP 100 8 bit data acquisition board (12) (Acqiris SA, Plan-les-Ouates, Switzerland), which provided a sampling rate of 1 GS/s. The mass spectra obtained from each laser pulse were summed “on board” at a rate of 174 spectra per second using the hardware accumulation facility. It was also possible to record a single mass spectrum after a single laser pulse.

A diode-pumped Nd:YAG laser with a 3rd harmonic generator (355 nm) was used. The laser pulse duration was 0.37 ns, the pulse energy was 40 μJ, and the pulse repetition rate could be adjusted from 1 to 300 Hz. To improve the energy stability, the laser was placed in a thermostat with a temperature accuracy of ±0.2 °C. At the repetition rate of 174 Hz, the 1 min stability was 0.1–0.3% RMS and 0.6–1.2% peak to peak. The laser radiation parameters were measured using a Gentec Maestro instrument with an XLPF12-3S-H2-D0 detector.

The laser energy distribution in the focal plane was close to Gaussian. To estimate/determine the parameters of the Gaussian distribution, we measured the size of the spots of visible laser damage on the surface of a polished silicon wafer at two values of the laser energy. The focal spot radius was estimated to be 0.046 mm at the 1/e level. An image of the laser damage on the wafer surface in both the static and scanning modes is shown in Figure 1b.

The scanning frame on the gold plate surface was 0.4x0.45 mm. The frame consisted of 25 lines with a step of 0.019 mm. The time for a single frame pass was 1 s. During a single pass, the laser beam covered approximately 15% of the frame area. The laser was not synchronized with the scanner; therefore, the alignment of the radiation spots was random during successive passes of the frame.

A CCD camera (11) was used to observe the target surface and find the plasma generation threshold. In front of the camera lens, a filter suppressing the scattered laser radiation was installed. The appearance of plasma was determined visually by the appearance of intense visible light emission, which was recorded using the CCD camera. 

To estimate the size of the nanoparticles, the products of a gold plate ablation were deposited on a piece of polished silicon wafer. It was installed in front of the grid (3) on a special holder placed at a distance of 5 mm from the gold plate (2). For the gold deposition, laser beam scanning was used and the laser pulse energy was 9 μJ. The value exceeded the plasma formation threshold on the gold surface.

The emission of both positive and negative Au ions depends not only on the laser fluence, but also on the condition of the metal surface, which changes upon laser irradiation. This effect is associated with the formation and modification of nanostructures on the metal surface exposed to the laser radiation, which effects the thermal and optical properties of gold [17,18,19]. To provide reproducible initial conditions, every measurement began with the ablation of the metal surface using laser radiation with an energy of 10 μJ (fluence of 140 mJ/cm^2^). After that, the required laser energy was set using the attenuator, and the mass spectra were recorded. To measure the dependences of the ion yield on the laser pulse energy, the procedure was repeated for each new value of the energy. The spectra were recorded in two regimes: (1) registration of a single spectrum for each laser pulse with a laser repetition rate of 10 Hz, and (2) use of hardware accumulation with summation of the accumulated spectra for 100 s with a laser repetition rate of 174 Hz. During this time interval, 17,400 mass spectra were recorded.

### 2.2. Scanning Electron Microscopy

For the analysis of the silicon wafer surface with deposited gold nanoparticles, a TESCAN SOLARIS scanning electron microscope (SEM; TESCAN ORSAY HOLDING, Brno-Kohoutovice, Czech Republic) was used.

### 2.3. Materials

Gold with a purity of 99.99% was purchased from Sberbank (Moscow, Russia). In the experiments with gold nanoparticle deposition, we used commercially available microelectronics grade wafers of Sb-doped n-type polished Si (100) with a resistivity of 0.01 Ohm∙cm (ELMA, Zelenograd Moscow, Russia).

## 3. Results

### 3.1. Mass Spectrometry

The measurements were carried out in both static and scanning modes, as described earlier. Figure 2 shows the negative Au ion and electron signals measured for every single successive mass spectrum recorded after a single laser pulse. During the experiment, the laser radiation energy increased stepwise, as is shown in Figure 2. As can be seen in the figure, both the negative ion and the electron signals strongly fluctuate, while the laser radiation energy and its spatial distribution do not change. Similar dependences are also observed for the positive ions. For both the scanning and static modes, the ion signals behave similarly, except for the difference in the magnitude. As can be seen in the figure, the amplitude of the ion and electron signal spikes, as well as the frequency of occurrence, increase with increasing laser radiation energy.

Figure 3 shows the mass spectra and the plots of the dependence of the ion signal of the positive Au ions on the number of pulses obtained both in scanning mode and in static mode (when the surface is irradiated at a fixed point). The mass spectrum consists of the peak of the Au^+^ ion and the peaks of the charged clusters (Au_2_^+^, Au_3_^+^, and Au_4_^+^). In the spectrum of the positive ions, minor ion peaks of silver (_107_Ag^+^ and _109_Ag^+^), iron (_56_Fe^+^), copper (_63_Cu^+^ and _65_Cu^+^), and alkali metals are also observed, as shown in the inset to Figure 3. The concentration of these elements in a gold sample with a purity of 99.99 does not exceed 10^−2^%. The mass spectra obtained both in scanning mode and in static mode at the same laser radiation energy and the ratio of the peaks of Au ions and charged clusters is almost the same, as is the composition of the impurities. However, the dependence of the ion signal on the number of laser radiation pulses (shown in Figure 3) is completely different. As can be seen in the figure, in scanning mode, the ion signal drops by approximately four times after a hundred-fold passage of the scanning frame. In static mode, a short surge of the ion signal is also observed within approximately the first 500 laser pulses (3 cycles of accumulation), and then the signal reaches an almost constant value with minor fluctuations. The difference between static mode and scanning mode should be noted. In static mode, every next laser pulse hits the same spot, while in scanning mode, every next laser pulse hits the adjacent area inside the scanning frame on the gold surface. For this reason, a much faster decreasing signal is observed in static mode in comparison to scanning mode. The maximal values of both signals are of the same order of magnitude.

The mass spectra of the negative ions differ only in the absence of peaks of silver, copper, iron, and alkali metals. The dependences of the signals of the negative Au ions on the number of scans are similar to those of the positive ions.

Figure 4a shows the dependences of the accumulated ion signal of the positive and negative Au ions on the laser pulse energy. The signal is calculated via the summation of the mass spectra recorded for 100 successive scanning frames (or for a time interval of 100 s). As a result, each point in the plot corresponds to the sum of 17,400 successive single mass spectra. The value of the integral of the ion signal is given in ADC counts. According to our estimates of the average amplitude of single-electron pulses from the ion detector, approximately three ADC counts correspond to one ion. The starting points of the plots correspond to 1000 counts, which means we register 1 ion per 50 laser pulses on average. The ending points of the plots in Figure 4 are limited by the ADC bit depth. The hatching highlights the area above the plasma formation threshold. Figure 4b shows the same dependences for the positive and negative Au_2_ and Au_3_ charged clusters.

As can be seen in the figure, all the dependences obtained in scanning mode can be approximated by a power function with almost the same exponent of 9 (std = 0.37). For the dependences obtained in static mode, the exponent is 15 (std = 0.59). 

The characteristic spikes of the ion and electron signals under exposure to pulsed laser radiation arise simultaneously. Figure 5a shows a record of the Au^−^ ion and electron signals for about 60 successive single mass spectra. Similar behavior on the part of the ion and electron signals was observed in all our experiments.

The difference between the ion and electron signals can be seen in Figure 5. The ion signal consists of individual spikes and zero signal intervals for some laser pulses between the spikes. The electron signal also contains spikes that correlate with the ion signal spikes. However, for the laser pulses that produce no ion signal, the electron signal does not drop to zero; rather, it remains at a certain level, which depends on the laser fluence. This level is shown by a dotted line in Figure 5a. The energy dependences of the average amplitude of the electron signal spikes for scanning mode and static mode (Figure 5b) are approximated by power functions with exponents of 1.97 and 2.43, respectively.

The results demonstrate that the formation of both positive and negative Au ions and charged Au clusters as well as the emission of electrons from the nanostructured gold surface exhibit stochastic behavior in the entire range of the laser radiation energy up to the plasma formation threshold (Figure 2). This means that not every successive laser pulse produces ions and electrons. The values of the ion and electron signals can significantly vary from pulse to pulse (see Figure 2 and Figure 5). The frequency of occurrence and the amplitude of the ion and electron signal spikes depend on the laser pulse energy.

By analyzing the train of Au ion signal spikes obtained while recording every single successive mass spectrum (see an example in Figure 2), one can obtain the average value of the ion signal spike amplitude *A* and the number of ion signal spikes *N* for a fixed number of laser pulses at a given energy *E*. The dependences *A*(*E*) and *N*(*E*), as determined for 500 successive laser pulses and presented in Figure 6, can be fitted by the power functions *A*(*E*) = *A*_0_*E^a^*^1^, *a*1 = 6.0 (std = 1.3) and *N*(*E*) = *N*_0_
*E^a^*^2^, *a*2 = 4.0 (std = 1.1), where *A*_0_ and *N*_0_ are constants. The average number of Au ions per laser pulse at a fixed laser pulse energy is proportional to the product *A*(*E*) × *N*(*E*).

The accumulation of 17,400 successive single mass spectra for a fixed value of the laser pulse energy allows us to obtain the dependence of the Au ion signal in the entire range of the laser pulse energy, which can be fitted well by a power function with the exponent *a* = 9 in scanning mode, as was mentioned above. Based on the available accuracy, the exponent *a* ≈ *a*1 + *a*2, where *a*1 and *a*2 are calculated while analyzing the successive single mass spectra obtained for the limited interval of the laser pulse energy.

The level of 1000 ADC counts for 17,400 laser pulses was taken as the threshold for ion appearance. Figure 4 shows that, in scanning mode, the thresholds for the appearance of positive and negative Au ions are 0.8 and 0.75 μJ, respectively (∼12 mJ/cm^2^), which is approximately 10 times lower than the plasma formation threshold. This value is close to the value of 10 mJ/cm^2^ obtained for a nanostructured gold surface [6]. In static mode, the threshold for the appearance of ions is significantly higher. It is 2.3 μJ (34 mJ/cm^2^) for both positive and negative ions, although it remains below the plasma formation threshold.

The pulsed laser irradiation of the gold surface with fluence above the plasma formation threshold makes the surface substantially rough, which complicates the particle size distribution measurement. In order to reveal the connection between the degradation of the Au ion emissivity under exposure to successive laser pulses and the change in the composition of the nanoparticles on an ion-emitting surface, we deposited nanoparticles on a piece of polished silicon wafer via the laser ablation of a gold plate, as described above in Section 2. The deposition time was 3 s (about 500 laser pulses) at a laser pulse energy of 10 μJ (laser fluence of 140 mJ/cm^2^), which exceeds the plasma formation threshold. Next, the silicon wafer with deposited nanoparticles was installed instead of the gold plate (see Figure 1a) as an ion emitter. The mass spectra of the positive and negative ions were recorded in scanning mode for about 3 min at a laser pulse energy of 1.2 µJ.

The normalized dependences of the Au^+^ ion signal on the number of scanning frames passed for a laser pulse energy of 1.2 µJ are shown on Figure 7. As can be seen, the decay of the ion signal in the case of the nanostructured gold surface and in the case of the silicon surface with deposited Au nanoparticles is practically the same.

### 3.2. SEM Surface Images

The silicon wafer surface with deposited nanoparticles was examined using a scanning electron microscope. The images of two regions of the deposition are shown in Figure 8: the first of an intact area and the other of an area after 3 min of laser scanning. The histograms of the particle diameter distribution shown in Figure 8c demonstrate that the laser scanning changes the spectrum of the nanoparticle sizes. Mostly, the particles smaller than 5 nm have disappeared. The total number of the particles in the full-size 0.5 × 0.5 µm SEM image (see Appendix A) is reduced by 1320 pieces during the laser scanning (from 3834 to 2514 particles). This corresponds to a density of removed particles of 5 × 10^11^ cm^−2^. As the scanning area is 1.8 × 10^−3^ cm^2^, the number of removed particles is 9 × 10^8^. This value is approximately 3–4 orders of magnitude higher than the number of Au ions and charged clusters registered at the given energy.

### 3.3. Numerical Simulations of Gold Surface Laser Heating

In order to reveal the correlation of the obtained thresholds values with the temperature of the irradiated gold surface, numerical simulations of the laser heating of bulk gold were carried out. The parameters of the numerical model corresponded to the experimental conditions: a laser pulse duration of 0.37 ns and fluences of 12 mJ/cm^2^ and 34 mJ/cm^2^, which corresponded to the ion appearance thresholds in scanning mode and in static mode, respectively. In our simulations, we took into account the temperature dependences of such parameters as the reflection, absorption, heat conductivity, and specific heat of the substrate material. The corresponding data were extracted from the available literature.

As the laser pulse duration is much longer than the electron-phonon relaxation time in bulk gold [20], the electron and lattice temperatures were taken to be equal. The optical penetration depth in gold at a wavelength of 355 nm is much less than 1 µm and the following condition holds: *d*
$\gg \sqrt{\chi \tau }$, where χ is the thermal diffusivity of the substrate, *d* is the laser spot diameter, and τ is the laser pulse duration. Thus, the problem of surface heating under pulsed laser radiation in our conditions can be considered to be one-dimensional.

We used the numerical model for non-stationary heat conduction with a distributed external heat source. The external heat source was distributed in the depth of the substrate according to the Buger–Lambert–Beer law, while the power temporal profile of the external heat source was defined by the Gaussian function with a width of 0.37 ns at the 1/e level. The dependences of the heat capacity and the heat conductivity on the temperature were taken from [21,22]. The optical parameters of gold, namely the penetration depth and reflectivity, were defined by the complex dielectric function (DF). In the UV and visible spectral range, the electronic interband transitions significantly affect the optical response, and the Drude free electron model fails to describe experimental data [23]. For this reason, the use of an approach suggested by Ujihara [24] for the thermal dependence of the DF is no longer valid. At a wavelength of 355 nm, the reflectivity and the optical penetration depth are defined mainly by the imaginary part of the DF. Experimental data describing the behavior of DF at temperatures up to 770 K are given in [25] and up to 800 K in [26]. As is noted in [27], the discrepancy in the experimental data concerning the DF is associated with the sample structure and surface condition. Variations in the DF at a wavelength 355 nm with the temperature [25,26] and from sample to sample [26,27] do not lead to significant differences in the maximal surface temperatures calculated for both threshold values of laser fluence: from 445 to 455 K for 12 mJ/cm^2^ and from 720 to 750 K for 34 mJ/cm^2^. In both cases, the maximal surface temperature is significantly below the melting point of bulk gold. However, changes in the surface structure begin at a temperature well below the melting point of bulk gold. It has been shown [28] that the static heating of micron-sized gold particles to a temperature of more than 500 K leads to the formation of layered nanostructures on their surface with a characteristic size up to 2 nm.

In order to compare the conditions at the surface of both substrates, namely gold and silicon, for the experiment described in Section 3.1, Figure 7, we calculated the maximal surface temperature under laser heating with a pulse energy of 1.2 µJ. The optical parameters for silicon were taken from [29], while the thermophysical parameters were taken from [30,31]. The calculated value for the bulk silicon substrate does not exceed 700 K, and it is less than 520 K for the bulk gold.

## 4. Discussion

The exposure of a nanostructured gold surface to pulsed laser radiation is accompanied by at least two physical processes. They are the process of formation of positive and negative ions of gold atoms and clusters and the process of rearrangement of the nanostructured surface of gold. The process of surface restructuring manifests itself as a change in the size of the nanoparticles. As can be seen in Figure 8, after exposure to laser radiation, the number of nanoparticles with sizes of less than 5 nm decreases and the remaining particles become somewhat enlarged. This process is accompanied by a strong decrease in the yield of all the types of ions desorbed from the surface. Note that the change during laser irradiation in the emission of ions from the gold surface preliminarily structured by laser radiation was observed earlier, although the temporal dynamics of the process have not been studied [11].

The ion signal dependences for static mode and for scanning mode presented in Figure 4 are attributed to different processes. In scanning mode, the ion signal corresponds to the overall number of gold nanoparticles produced on the surface under the preceding laser processing at elevated energy. In static mode, the nanoparticles previously produced under laser processing are rapidly destroyed, which gives the short surge of the ion signal. The stationary value of the ion signal is apparently the result of the competition between the destruction and generation of gold nanoparticles under laser exposure.

A change in the size of the nanoparticles also leads to an increase in the ion formation threshold. As can be seen in Figure 4, the ion formation threshold increases from 12 mJ/cm^2^ for the scanning mode to 34 mJ/cm^2^ for the static mode. These values correspond to the different initial size distributions of the nanoparticles. In scanning mode, the threshold is determined for a surface containing a significant number of nanoparticles of less than 5 nm in size. In static mode, after exposure to hundreds of laser pulses at a fixed point, such nanoparticles are removed and we deal with a surface containing larger nanoparticles. However, in both modes, the ion formation threshold is significantly lower than the plasma formation threshold on the surface of nanostructured gold. Moreover, as shown by our calculations, the temperature of the gold surface at laser fluence corresponding to the ion formation threshold was 455 K and 750 K for the scanning and static modes, respectively. Both values are much lower than the melting temperature of bulk gold (1337 K). The situation was similar for the surface of the silicon plate, where the temperature did not exceed 700 K in our experiment.

There are at least two mechanisms that lead to a change in the sizes of particles. It is well known that the impact of nanosecond or femtosecond radiation on colloidal solutions of gold nanoparticles leads to reshaping at moderate laser radiation intensities [32,33]. The process of particle size reduction in a solution is limited by the thermal conductivity and phase transition of the solvent. When nanoparticles are heated in a vacuum, such thermal confinement does not work, at least not for the particles weakly bound to the surface of the bulk material, and this can result in the complete destruction (evaporation) of the particles. The reverse process is also possible—coalescence of the nanoparticles, leading to the coarsening of the particles [34,35]. Both processes take place in our experiment and lead to the observed decrease in the number of nanoparticles smaller than 5 nm.

In the case of a nanostructured gold surface exposed to pulsed laser radiation, the process of ion formation can hardly be explained using the commonly considered mechanisms—ionization on the surface, photoionization of metal vapors desorbed by laser radiation, and ionization of vapors in laser-induced plasma—since the laser fluence used in our experiment is insufficient to trigger these processes. Ion appearance in that case is a very rare event. The mean number of gold ions per nanoparticle within the laser spot varies from approximately 10^−8^ to 10^−4^ per one laser pulse.

An unexpected result of our experiment is the practically constant ratio of the yield of positive and negative Au ions, as well as positive and negative charged clusters, in the entire range of the laser energy (Figure 4). It is well known that the emission of positive ions from laser plasma always exceeds the emission of negative ones, which was shown by direct mass spectroscopic measurements [36]. However, in our experiments, the negative ion yield exceeds the positive ion yield by a factor of approximately two in the scanning mode, while the positive and negative ion yields are almost equal in the static mode.

Another unexpected result is the fact that the dependences of the yield of both the positive and negative Au ions and charged Au clusters on the laser energy are the same. It is obvious that this process cannot be associated with surface ionization, as for any noticeable yield of positive ions due to the surface ionization of Au atoms on the gold surface, a high surface temperature is required. This temperature depends on the difference between the work function (4.8 eV) and the ionization potential (9.22 eV), and it is not reachable in the experiment. In addition, if we relate the laser fluence and the surface temperature of gold, then with a change in the fluence, the dependences of the yield of positive and negative ions should differ according to the Saha–Langmuir formula [37].

A possible explanation lies in the already made assumption regarding the emission of ions during the fragmentation of a nanoparticle under pulsed laser excitation. A nanoparticle that is weakly bound to the surface or located above the surface can be destroyed due to two possible effects. The first of these is the Coulomb explosion [38]. Under our experimental conditions, this is an unlikely process, as the laser pulse duration is much longer than the characteristic electron–phonon relaxation time in gold. Obviously, under our conditions, one should consider the destruction of a nanoparticle to be a result of heating. Since the duration of the laser pulse significantly exceeds the electron–phonon relaxation time, the particle can repeatedly go through absorption and relaxation cycles during the pulse, which leads to its heating. If a sufficiently high amount of energy is absorbed, the destruction of the particle and the formation of smaller clusters can occur. This process is accompanied by the thermal emission of electrons and can lead to the formation of positively charged atoms and clusters. At high temperature, decreasing the size of a charged particle because of the evaporation of neutral atoms from its surface can lead to the destruction of the particle by Coulomb forces, as happens when a liquid is sprayed in an electric field [39]. However, this cannot explain the formation of negative clusters of Au_n_^-^ and the irregular spike-like behavior of the ion signal.

We speculate that the signal spikes observed result from gold nanoparticles destruction. At higher laser energy, the ion signal increases due to the increase in the number of nanoparticles destroyed under a single laser pulse. As shown above (Figure 8), laser irradiation reduces the number of nanoparticles smaller than 5 nm. The mechanism of light absorption by particles in this size range varies from absorption by electrons in the metal conduction band to the excitation of electronic transitions in a single Au atom [40].

The excitation and ionization of individual gold atoms by means of laser radiation is unlikely, as direct three-photon ionization without the excitation of the intermediate levels requires much higher laser radiation intensities. Stepwise photoionization requires the resonant excitation of the intermediate energy level, followed by the ionization of the excited atom by a quantum of a different wavelength, as it is implemented in laser isotope separation methods [41] or similar REMPI processes [42]. This requires two different lasers, and at least one of them must be tunable.

The results of [43] indirectly confirm that particles smaller than 5 nm are involved in the ionization process. There was reported no characteristic plasmon resonance peak in the spectral dependence of the Au ion yield during the laser ionization of the nanoparticles desorbed by laser radiation from a water jet in a vacuum. The absence of a plasmon resonance peak in the absorption spectrum is characteristic of nanoparticles smaller than 5 nm [44]. The laser wavelength (335 nm) is much less than the plasmon resonance wavelength characteristic of larger gold particles in our experiment. In other words, plasmon resonance excitation is not involved in the process of ion formation.

The absorption of light by a nanoparticle (cluster) consisting of three or more atoms differs from the absorption by a single atom. An N-atom nanoparticle has *f* = 3*N* − 6 degrees of freedom (in the case of a linear arrangement of atoms, *f* = 3*N* − 5). As the vibrational energy increases, the density of the vibrational states can be estimated in a rough approximation using the following formula [45]:(1)$$P=\frac{G_{0}^{f-1}}{\left(f-1\right)!\;\prod_{i}\nu_{i}},$$
where *P* is the density of levels per unit of energy, *ν_i_* is the vibrational frequency, and *G*_0_ is the vibrational energy in the same units of energy as the vibrational frequencies.

The power-law dependence of the density of the energy states on the vibrational energy can lead to the formation of a vibrational quasi-continuum above a certain vibrational energy, namely in the region where the anharmonic interaction of nearby vibrational states exceeds the energy difference between these states. In the quasi-continuum of vibrational levels, in a time interval inversely proportional to the anharmonic interaction strength, the redistribution of vibrational energy between different vibrational modes can occur. The energy relaxation inside the nanoparticle is similar to the intramolecular relaxation in large molecules.

This can, in turn, lead to the statistically uniform distribution of the vibrational energy over the vibrational degrees of freedom. In addition, intramolecular relaxation leads to the broadening of the vibrational absorption bands of electronic transitions. In other words, in the case of electronic transitions, due to the vibrational quasi-continuum, broad absorption bands are formed, which makes it possible to successively resonantly excite high-lying electronic states by means of laser radiation.

With a decrease in the size of a nanoparticle under successive exposure to laser pulses, the structure of the electronic transitions is rearranged and the mechanism of broadening of the transition bands due to the vibrational quasi-continuum is turned off, which can lead to the formation of narrow absorption bands and, possibly, an increase in absorption at a wavelength of 355 nm for some particle in the small size range. This can provide the effective excitation of the nanoparticle, which can lead to its fragmentation or ionization. In polyatomic clusters, the total value of the stored vibrational energy upon laser excitation can exceed the dissociation energy and the ionization energy. According to RRKM theory [46], the rate of dissociation of overexcited clusters into neutral fragments can turn out to be quite low, even if the excitation energy is noticeably higher than the dissociation threshold, as the statistical probability of energy concentration in the broken interatomic bond and the formation of a transition state from which the cluster can decay along one of the possible channels is small. The decay rates through various channels must increase with the reserve of vibrational energy.

One of the most probable channels for the formation of ions in this case may be the detachment of an electron with the formation of a positively charged nanoparticle Au_n_^+^. The ionization potential of a nanoparticle is somewhat less than the ionization potential of an Au atom (9.1 eV) and depends on its size, which facilitates ionization. However, this does not explain the formation of negative ions or the almost constant ratio of positive and negative ion signals, unless we assume that the emitted free electrons will inevitably be captured by neutral nanoparticles. The formation of negative ions can indeed be associated with the capture of electrons emitted from a particle or from the gold surface by neutral clusters. As can be seen in Figure 5b, the dependence of the electron signal on the laser fluence is approximately quadratic, while similar dependences for ions and charged clusters are described by a power function with an exponent of 9 (see Figure 4), which indicates that the process of negative ion formation is weakly related to the presence of free electrons. The electron signal and the signal of the ions and charged clusters are about the same order of magnitude at laser pulse energy near 1 µJ.

We believe that the excitation of a nanoparticle through a quasi-continuum of electronic-vibrational states leads to another channel for the formation of ions—the dissociation of a neutral nanoparticle into two oppositely charged particles of smaller size. This process is energetically less favorable than the detachment of an electron. The energy of two ions formed from an Au_2_ dimer can be estimated as a sum of the dissociation and ionization energies minus the electron affinity. This value is near 15 eV, and it decreases as the particle size increases. The energetic unfavorability of such a process can be compensated for by the low rate of dissociation of a nanoparticle into neutral fragments, which, according to the RRKM theory, can slowly increase with an increase in the excitation energy, in comparison with the rate of excitation of electronic states. A similar ionization mechanism is discussed in [47]. In that study, C_60_ molecules were excited by pulsed IR radiation with a wavelength of 20 µm. Stepwise excitation through a quasi-continuum can lead to the absorption of sufficient energy to ionize the polyatomic molecule.

The extremely fast ion yield dependencies on the laser energy, as described by the power law with the exponent x = 9 for the scanning mode and x = 15 for the static mode, can also be associated with the processes for the excitation of nanoparticles. As already noted, an increase in the vibrational energy of a nanoparticle leads to a power law increase in the level density and the formation of a quasi-continuum. This increases the probability of nanoparticle excitation by means of laser radiation and, as a result, increases the ion yield with increasing laser energy.

The difference in the slopes of the ion yield dependences for the scanning mode and static mode is due to the nanoparticles that are present on the substrate surface during the entire measurement process in the scanning mode. In the static mode, the nanoparticles, which were previously prepared by irradiating the surface with laser fluence above the plasma formation threshold, “burn out” in the first few tens of laser pulses, which results in fast signal decay (Figure 3). To reach the same signal level as in the scanning mode, we needed to increase the laser pulse energy by a factor of approximately 3. We believe that in this case, new nanoparticles are generated at higher laser fluence during the measurement process, which leads to faster signal increase. To restore them, it is necessary to increase the laser fluence to a level that provides a change in the surface. This leads to the increase in the ion formation threshold observed in the case of accumulating a number of mass spectra.

## 5. Conclusions

The formation of ions on a nanostructured gold surface under pulsed laser radiation in a vacuum is a complex process comprising a number of physical phenomena. The process of ion formation studied in this work under laser fluence below the plasma formation threshold on a nanostructured gold surface gives rise to a rather small number of ions. In our experiments, the mean number of produced ions varied from 0.1 to several hundred per laser pulse. The observed ion and electron signals looked like a train of irregular spikes with fluctuating amplitudes and frequencies of occurrence. This effect is associated with the heating of nanoparticles under exposure to a laser pulse with a pulse duration much longer than the electron–phonon relaxation time in Au particles. We believe that particles less than 5 nm in size are mainly involved in ion formation. The excitation of such particles in a laser field with a fixed wavelength is facilitated by the formation of a quasi-continuum, and the energy absorbed by a nanoparticle can significantly exceed the dissociation and ionization energies. The above considered mechanisms of ion formation, such as surface ionization, photoionization with the detachment of an electron from a nanoparticle with the formation of a positive ion, and fragmentation of a particle due to thermal emission of electrons and its evaporation, do not explain the fact that the ratio of the numbers of positive and negative Au ions and clusters exhibits low variability in the entire range of the laser energy. We suppose that in our experiments, an energetically less favorable process was realized, i.e., the dissociation of nanoparticles strongly excited in a laser field into two oppositely charged ions. The experimental power-law dependences of the ion yield from the nanostructured gold surface on the laser energy could also be related to the features of the excitation of nanoparticles in the laser field.

We believe that the spike-like ion and electron signals behavior and the practically constant ratio of the numbers of positive and negative Au ions are determined by the peculiar features of light absorption by a nanoparticle. Since the structure of the electronic transitions in a nanoparticle is determined to a greater extent by size effects than by the structure of the electronic transitions of its constituent atoms, the observed effects should also manifest when laser radiation is applied to nanoparticles of other metals.

## Figures and Tables

**Figure 1 nanomaterials-13-00600-f001:**
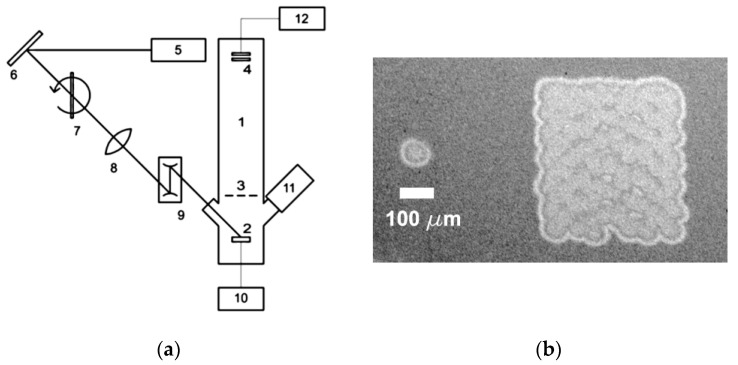
(**a**) Experimental setup: 1—time-of-flight mass spectrometer, 2—gold plate, 3—ion source grid, 4—ion detector, 5—laser, 6—mirror, 7—attenuator, 8—lens, 9—scanner, 10—high voltage power supply, 11—CCD camera, and 12—data acquisition board. (**b**) The laser damage spots on the silicon wafer surface in static mode (left) and scanning mode (right).

**Figure 2 nanomaterials-13-00600-f002:**
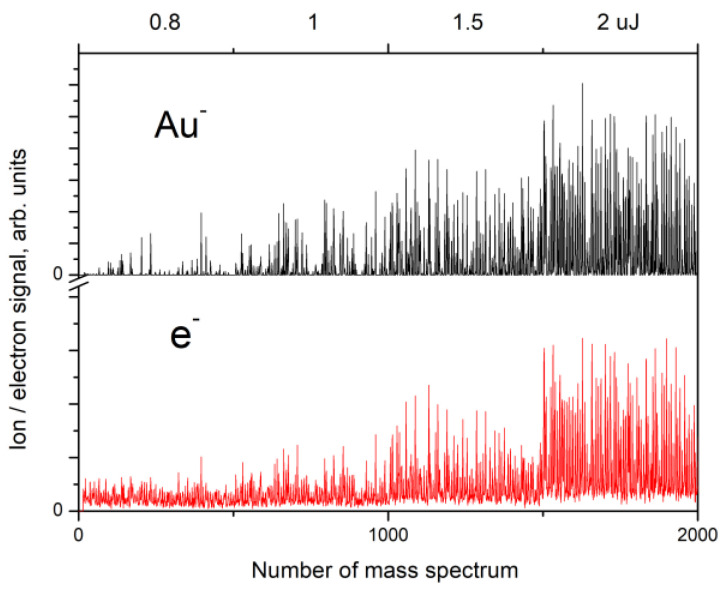
Au^−^ ion and electron signals for a sequence of single mass spectra recorded after every single laser pulse. The laser radiation energy increased stepwise (values and intervals are shown at the top). Scanning mode.

**Figure 3 nanomaterials-13-00600-f003:**
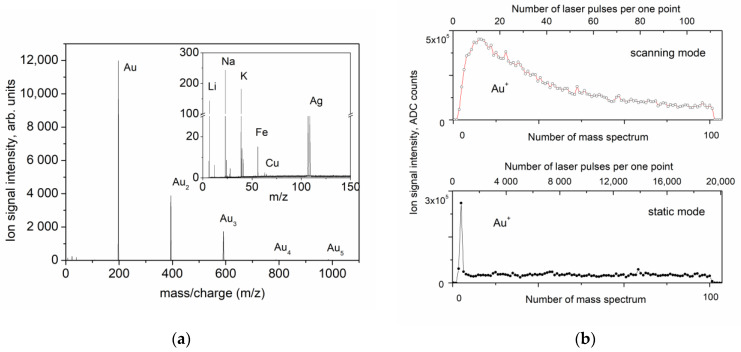
(**a**) Mass spectrum of the positive ions in static mode at a laser radiation energy of 2.7 μJ. Mass spectrum of the major impurities is presented in the inset. (**b**) Evolution of the Au^+^ ion signal under successive exposure to laser pulses in scanning mode (top) and in static mode (bottom). Each point represents the sum of 174 successive mass spectrum records. The additional axis on the top of each diagram shows the number of laser shots per one point on the gold surface.

**Figure 4 nanomaterials-13-00600-f004:**
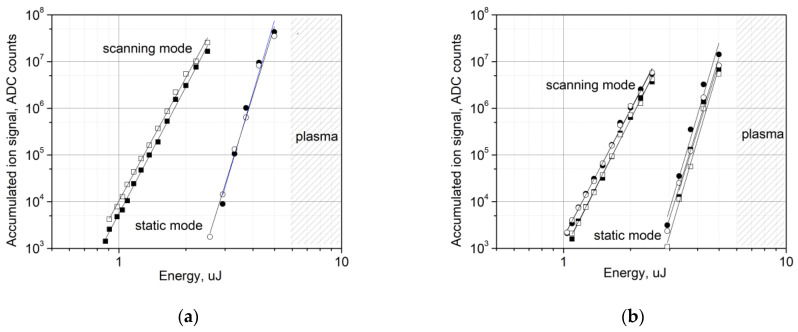
(**a**) The dependences of the accumulated ion signal vs. laser pulse energy for the positive (filled symbols) and negative (open symbols) Au ions as measured in scanning mode and in static mode. (**b**) The dependences of the accumulated ion signal vs. laser pulse energy for the positive (filled symbols) and negative (open symbols) Au_2_ (circles) and Au_3_ (squares) charged clusters as measured in scanning mode and in static mode. Each point on the plot corresponds to 17,400 accumulated successive single mass spectra (100 s time interval). An approximation of the experimental data by a power function is shown by the solid lines. The hatching highlights the region of plasma formation.

**Figure 5 nanomaterials-13-00600-f005:**
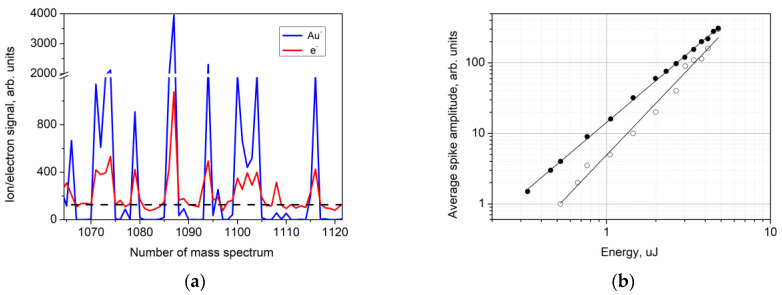
(**a**) A record of the Au^−^ ion signal (blue) and e^-^ electron signal (red) for successive single mass spectra. (**b**) Average amplitude of the electron signal spikes vs. the laser pulse energy in scanning mode (filled circles) and in static mode (open circles).

**Figure 6 nanomaterials-13-00600-f006:**
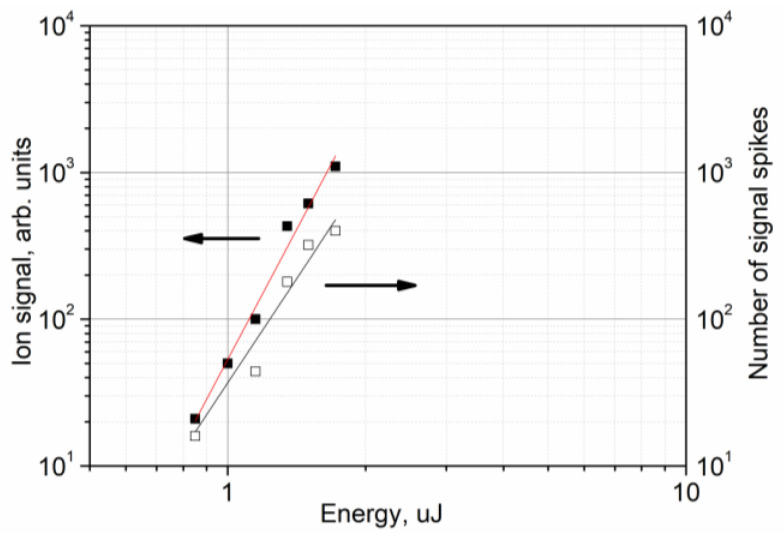
The dependences of the negative Au ion signal spike amplitude (filled squares) and the number of signal spikes (open squares) on the laser pulse energy. Scanning mode. The data were obtained for 500 successive laser pulses.

**Figure 7 nanomaterials-13-00600-f007:**
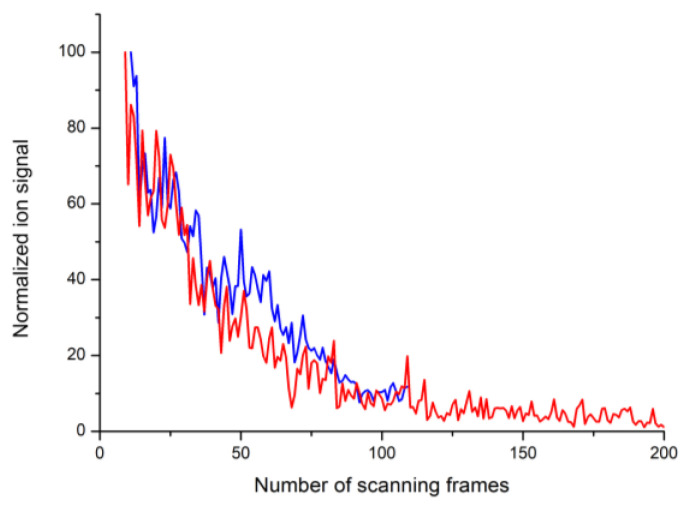
Evolution of the Au^+^ ion signal under exposure to successive laser pulses vs. the number of scanning frames passed for the nanostructured gold (red) and for the surface of the silicon wafer with nanoparticles deposited via laser ablation (blue). The laser pulse energy was 1.2 μJ.

**Figure 8 nanomaterials-13-00600-f008:**
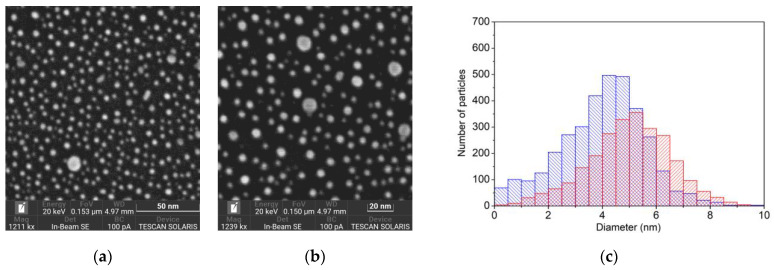
(**a**) A fragment (153 × 153 nm) of the SEM image of the intact silicon wafer surface with deposited Au nanoparticles. (**b**) A fragment (150 × 150 nm) of the SEM image of the silicon wafer surface with deposited Au nanoparticles after 3 min of scanning with pulsed laser radiation (laser pulse energy of 1.2 μJ). (**c**) Histogram of the nanoparticle diameter distribution before (blue) and after (red) laser exposure. The Au nanoparticles were deposited by about 500 laser pulses at an ablating laser pulse energy of 10 μJ.

## Data Availability

The data presented in this study are available in the article and Appendix A.

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
