# Peer review of "Laser-Induced Ion Formation and Electron Emission from a Nanostructured Gold Surface at Laser Fluence below the Threshold for Plasma Formation"

_nanomaterials, 2023, doi:10.3390/nano13030600_

Round 1

Reviewer 1 Report

The present work deals with laser ablation of gold surfaces using nanosecond laser pulses of 355 nm wavelength in vacuum. The main take-home-message seems to be the fact that laser induced gold ion and cluster generation (positive as well as negative), being emitted from gold surfaces, occurs at laser fluence below the threshold of plasma formation while using 355 nm laser wavelength. The mechanism is discussed with a special emphasis on the role of Au nanoparticles with sizes below 5 nm.

The topic may be of potential interest to the readers of Nanomaterials, however, the discussion section of the manuscript is poor and should be rewritten. In the current version, the discussion section is unclear.

Minor mistake is on page 9, line 329: section "3.3" instead of "3.2" should be written.

On page 12, line 485, Figure 5b is addressed, however, no quadratic vs. power function of 9 is obvious in Figure 5b. This is one of the confusing parts of discussion section.

Furthermore, Figures have to be enlarged so that all numbers and letters written there are readable easily. In Figure 8c, page 9, it seems that the deposited Au nanoparticles on Si wafer are increasing in size when compared before and after the laser impact (blue and red histograms, respectively). This is exactly the oposite fact than the one which was observed in the literature and discussed by the authors in discussion section. However, the discrepancy might be caused by the puzzling discussion section in the present manuscript. 

Based on the above mentioned, I would require major revisions

Reviewer 2 Report

nanomaterials-2144354

This paper is interesting and well written and should be published with minor revision.  There are some minor editorial suggestions listed below.

The most interesting part of this manuscript is the observation of equal numbers of positive and negative gold ions.  The explanation is also novel.

Editorial corrections:

Line 187: similarly

Line 281-283: We speculate that the signal spikes observed are the result of gold nanoparticle destruction. At…….nanoparticles destroyed under….pulse.

Line 500: In that study…

Line 531: We believe that particles less than 5 um in size are mainly involved in the ion formation.

Reviewer 3 Report

The work is aimed of the topical issue of studying the mechanisms of emission of positive and negative ions from the surface of gold under the influence of pulsed laser radiation. A feature of this process is the energy below the laser ablation threshold. The obtained data make a significant contribution to the study of the issue, but there are a number of questions and comments for the authors.

1.    Obviously, the static and scanning regimes should lead to the different degrees of the target degradation. Therefore, a more unambiguous description of the differences between the static mode and the scan mode is required (for example, the number of laser pulses hitting on the same target surface). Otherwise, the obtained data are poorly interpreted, in particular, the dynamic curves in Figure 3.

2.     Line 283. Probably the link to Figure 6 and not Figure 8?

3.     The authors rightly believe that the plasmon absorption for 5 nm particles is insignificant. However, according to the Figure 8, larger nanoparticles are also observed on the substrate surface, up to 8–10 nm, for similar particle size, plasmon absorption is sufficiently effective. Moreover, nanoparticles have an energy structure that differs from the energy structure of clusters. All these differences are quite easily visualized using absorption spectroscopy in the UV-vis range, and in the case of clusters, also luminescence spectroscopy. It is necessary to present the reflection/absorption spectra for the studied nanostructured samples. This will justify the assumptions and conclusions made in the article.

4.     The absorption of laser radiation by the silicone substrate during laser action on gold nanoparticles wasn’t taken into account. Nevertheless, photons with 355 nm wavelength effectively create free electrons near the silicon surface, the relaxation of which can lead to quite efficient heating of the surface. Taking into account the fact that the melting temperature of gold nanoparticles decreases with decreasing size, the substrate temperature reached during the laser pulse can facilitate efficient evaporation of nanoparticles smaller than 5 nm. Has the maximum temperature of the silicon substrate been estimated under the influence of laser radiation?

Reviewer 4 Report

This paper reports the laser induced ion formation and electron emission from a nanostructured gold surface. The authors claim that the ratio of the signals of positive and negative ions is constant over the entire range of the laser radiation energies. This work can be generally interesting for their potential application electron/ion source. However, the paper introduction is not well focused so that I cannot understand what is the merit of this paper. In addition, the paper structure and description including figures quality are not well presented. For example, in section “3.2. Numerical simulations of gold surface laser heating”, I can find any curves or figures  to present the numerical results. In figure3b, I cannot find out the data for ion signal intensity, it is hard to compare the results for scanning and static modes of laser irradiations (ie arb. unit can be acceptable, but the data is must). As a result, I suggest reject of this paper in the current format.

Round 2

Reviewer 1 Report

I would like to thank to the authors that they took their time and enlarged discussion section substantially, included a clear statement of novelty in introductory part, improved the clarity of Figure 3b. I would recommend to increase the letters and numbers in figures so that for instance axes description is readable even after decrease of figures into the final format, but it is rather a formal issue.

 In my humble opinion, it is now publishable and I recommend it for publishing in Nanomaterials. 

Reviewer 3 Report

The authors have made all the necessary corrections. The work is recommended for publication in the journal Nanomaterials.

Reviewer 4 Report

Accepted.